# The potential of in situ cosmogenic $^{14}$CO in ice cores as a proxy for galactic cosmic ray flux variations

**Vasilii V. Petrenko**[1], **Segev BenZvi**[2], **Michael Dyonisius**[1,a], **Benjamin Hmiel**[1,b], **Andrew M. Smith**[3], **and Christo Buizert**[4]

[1]Department of Earth and Environmental Sciences, University of Rochester, Rochester, NY, USA
[2]Department of Physics and Astronomy, University of Rochester, Rochester, NY, USA
[3]Centre for Accelerator Science, Australian Nuclear Science and Technology Organization, Lucas Heights, NSW, Australia
[4]College of Earth, Ocean and Atmospheric Sciences, Oregon State University, Corvallis, OR, USA
[a]present address: Institute of Arctic and Alpine Research, University of Colorado Boulder, Boulder, CO, USA
[b]present address: Air Pollution Control Division, Colorado Department of Public Health and the Environment, Glendale, CO, USA

**Correspondence:** Vasilii V. Petrenko (vasilii.petrenko@rochester.edu)

**Abstract.** TS1 Galactic cosmic rays (GCRs) interact with matter in the atmosphere and at the surface of the Earth to produce a range of cosmogenic nuclides. Measurements of cosmogenic nuclides produced in surface rocks have been used to study past land ice extent as well as to estimate erosion rates. Because the GCR flux reaching the Earth is modulated by magnetic fields (solar and Earth's), records of cosmogenic nuclides produced in the atmosphere have also been used for studies of past solar activity. Studies utilizing cosmogenic nuclides assume that the GCR flux is constant in time, but this assumption may be uncertain by 30 % or more. Here we propose that measurements of $^{14}$C of carbon monoxide ($^{14}$CO) in ice cores at low-accumulation sites can be used as a proxy for variations in GCR flux on timescales of several thousand years. At low-accumulation ice core sites, $^{14}$CO in ice below the firn zone originates almost entirely from in situ cosmogenic production by deep-penetrating secondary cosmic ray muons. The flux of such muons is almost insensitive to solar and geomagnetic variations and depends only on the primary GCR flux intensity. We use an empirically constrained model of in situ cosmogenic $^{14}$CO production in ice in combination with a statistical analysis to explore the sensitivity of ice core $^{14}$CO measurements at Dome C, Antarctica, to variations in the GCR flux over the past $\approx 7000$ years. We find that Dome C $^{14}$CO measurements would be able to detect a linear change of 4 % over 7 ka, a step increase of 4 % at 3.5 ka or a transient 100-year spike of 250 % at 3.5 ka at the $3\sigma$ significance level. The ice core $^{14}$CO proxy therefore appears promising for the purpose of providing a high-precision test of the assumption of GCR flux constancy over the Holocene.

## 1 Introduction

The galactic cosmic ray (GCR) flux at Earth is modulated by both the geomagnetic and the heliospheric (solar) magnetic fields. The heliospheric magnetic field strength is linked to solar activity and solar irradiance (e.g., Wu et al., 2018b; Steinhilber et al., 2009), with irradiance being a key climate driver. This has enabled the use of records of past cosmogenic nuclide production rates for studies of past solar variability (e.g., Adolphi et al., 2014; Bard et al., 2000; Steinhilber et al., 2009; Usoskin et al., 2016; Usoskin, 2023). The two main nuclides that have been used for these studies are $^{14}$C (mainly from tree rings, which record atmospheric $^{14}$C/$^{12}$C ratio) and $^{10}$Be (from ice cores, which record the flux of $^{10}$Be at the snow deposition site). Measurements of cosmogenic nuclides have also been used extensively to study processes at the Earth's surface, such as the timing of glacial advance and retreat (e.g., Balco, 2020). In these ap-

plications, nuclides such as $^{10}$Be, $^{14}$C, $^{26}$Al and $^{36}$Cl that are produced in situ in surface rocks are of interest.

Cosmogenic nuclide-based reconstructions of past solar activity and ice extent have relied on the assumption that the GCR flux is constant in time (e.g., Balco, 2011; Gosse and Phillips, 2001; Muscheler, 2013). Measurements of cosmogenic radionuclides in meteorites provide arguably the strongest support for this assumption (e.g., Smith et al., 2019; Wieler et al., 2013). However, a number of important uncertainties are involved when interpreting these measurements, including meteoroid orbits, solar modulation of the GCR flux and break-up of meteoroids/fresh surface exposure on entry into the atmosphere. A review by Wieler et al. (2013) concluded that while overall the meteorite evidence indicates that the GCR flux is constant, this assumption is uncertain by 30 % or more. Measurements of cosmogenic nuclides in lunar rocks also indirectly indicate that the GCR flux could have been constant on million-year timescales, although there is still a confounding influence of solar modulation (e.g., Poluianov et al., 2018). Records of $^{10}$Be/$^{9}$Be ratios in oceanic sediments and iron–manganese crusts (Willenbring and von Blanckenburg, 2010) have also been used to argue that the GCR flux is approximately constant on million-year timescales (Wieler et al., 2013). However, this approach also involves multiple confounding factors, such as solar and geomagnetic modulation of the GCR flux and $^{10}$Be transport, deposition, and oceanic cycling. Results from studies that have used cosmogenic $^{14}$C and $^{10}$Be to examine past solar activity also assume that there were no large changes in the GCR flux in the past few millennia (e.g., Knudsen et al., 2009; Steinhilber et al., 2012; Wu et al., 2018a). However, again, inferences about the GCR flux from such records are complicated by solar and geomagnetic modulation (e.g., Knudsen et al., 2008), carbon cycle (for $^{14}$C, e.g., Muscheler et al., 2007), and transport and deposition effects (for $^{10}$Be, e.g., Field et al., 2006).

Theoretical considerations also generally support the assumption that the GCR flux is constant, though small anisotropies are expected due to the effect of the nearest sources of GCRs and the diffusive propagation of cosmic rays in the galaxy (Erlykin and Wolfendale, 2006; Blasi and Amato, 2012; Ahlers and Mertsch, 2015; Mertsch and Funk, 2015). At energies above 100 GeV, the GCR flux at Earth today is isotropic to within 1 part in 1000, with the residual anisotropy characterized by a dipole plus statistically significant components on angular scales as small as 5° (e.g., Abeysekara et al., 2019, and references therein). The observations indicate that cosmic ray transport is dominated by diffusion in galactic magnetic fields, which should dampen the contributions of spatial and temporal point sources of cosmic rays. Nevertheless, significant GCR flux variations are in principle possible even on sub-millennial timescales. For example, Melott et al. (2017) and Thomas et al. (2016) consider the terrestrial effects of a supernova 50 parsecs from Earth and estimate that the production rate of atmospheric muons could increase by up to several orders of magnitude depending on how accelerated GCRs propagate through nearby galactic magnetic fields. While the predictions of such models should be understood to represent the extreme upper limit of possible effects, a number of observations suggest that supernova explosions in our galactic neighborhood do produce measurable effects on the local properties of GCRs. Such observations include the part-per-mille dipole anisotropy in the cosmic ray flux above 1 TeV (e.g., Abeysekara et al., 2019; Ahlers and Mertsch, 2015; Blasi and Amato, 2012; Erlykin and Wolfendale, 2006), the fluxes of positrons and antiprotons above 20 GeV and heavy nuclei above 1 TeV (e.g., Kachelriess et al., 2015), and the measurements of $^{60}$Fe in ocean sediments (Wallner et al., 2016) and Antarctic snow (Koll et al., 2019). Thus, high-precision tests of GCR flux variations that are free of the confounding factors discussed above for meteorites and for cosmogenic $^{10}$Be and $^{14}$C produced in the atmosphere would be valuable.

## 2 Systematics of in situ cosmogenic $^{14}$CO in glacial ice

### 2.1 Overview of $^{14}$CO in glacial ice

We first provide an overview of the current understanding of the processes that control the abundance of in situ cosmogenic $^{14}$CO in glacial ice, which is needed to understand how the ice core $^{14}$CO proxy for GCR flux variations works. $^{14}$C in glacial ice originates from trapping of $^{14}$C-containing atmospheric gases such as carbon dioxide ($CO_2$), methane ($CH_4$) and carbon monoxide (CO) as well as from in situ cosmogenic production. In situ $^{14}$C is produced in glacial ice and firn via interactions of secondary cosmic ray neutrons and muons with $^{16}$O in the ice grains (Fig. 1a) (e.g., Lal et al., 1997; Petrenko et al., 2016; van der Kemp et al., 2002). Once produced, this $^{14}$C reacts rapidly to form predominantly $^{14}CO_2$ and $^{14}CO$, with a small amount of $^{14}CH_4$ and possibly other organics also being formed (e.g., Fang et al., 2021; Lal et al., 2000; Petrenko et al., 2013; van de Wal et al., 2007). $^{14}$C production rates are highest near the surface, where neutron-induced spallation of $^{16}$O is the main production mechanism. The neutron flux is attenuated rapidly with depth, however, and only affects the uppermost $\approx 20$ m of the firn (or uppermost $\approx 10$ m of solid ice) (e.g., Lal et al., 1987). Below these depths, production of $^{14}$C proceeds at lower rates and is dominated by negative muon capture as well as interactions with fast muons (Fig. 1b) (Petrenko et al., 2016; van der Kemp et al., 2002).

The concentration of in situ $^{14}$C in glacial ice at accumulation sites is controlled by the $^{14}$C production rates (site and depth-dependent), the snow accumulation rate and the retention of $^{14}$C in the firn. Sites at higher altitudes have less atmospheric shielding from cosmic rays, resulting in higher $^{14}$C production rates at the surface (e.g., Lifton et al., 2014). At sites with low accumulation rates, ice layers spend more

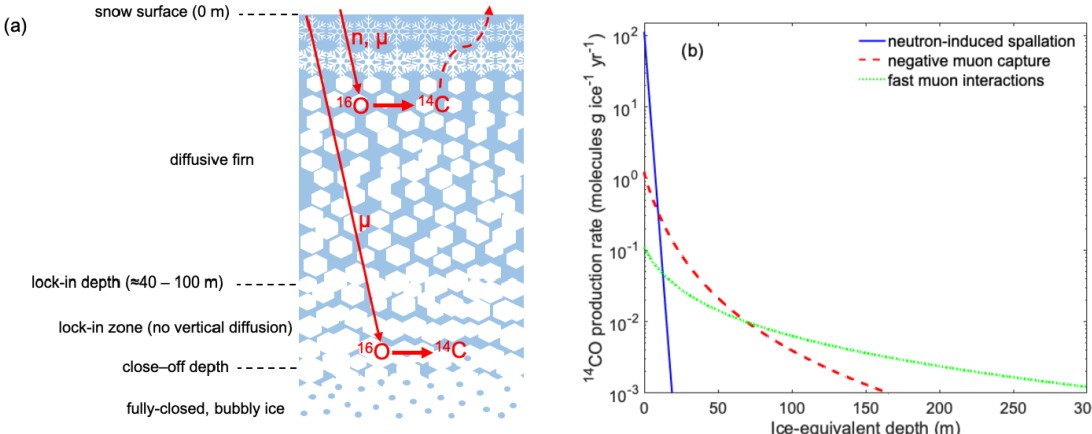

**Figure 1.** Introduction to production and loss of $^{14}$C in firn and ice. **(a)** Simplified schematic of the firn column, illustrating in situ cosmogenic production $^{14}$C by neutrons ($n$) and muons (μ) and loss in the upper, diffusive part of the firn as well as $^{14}$C production by muons below the lock-in depth where all of the $^{14}$C is retained. **(b)** $^{14}$CO production rates calculated as described in Sect. 2.2. Here $f_{\mu-} = 0.0667$ and $f_{\mu\mathrm{f}} = 0.0722$, which are mid-range values from the range constrained by Hmiel et al. (2023).

time at relatively shallower depths, allowing for more in situ $^{14}$C to be produced. Further, prior work has shown that most of the in situ cosmogenic $^{14}$C that is produced in the firn is rapidly lost to the atmosphere (de Jong et al., 2004; Hmiel et al., 2023; Petrenko et al., 2013). Because of this, the majority of the in situ cosmogenic $^{14}$C in glacial ice is from production below the firn zone. Of the $^{14}$C-containing gases in glacial ice, $^{14}$CO has the highest ratio of in situ cosmogenic to trapped atmospheric $^{14}$C. This is due to (1) atmospheric $^{14}$CO concentrations being lower than those for $^{14}$CO$_2$ or $^{14}$CH$_4$ (mainly because global mean mole fractions for CO ($\approx 80\,\mathrm{nmol\,mol^{-1}}$) are much lower than those for CO$_2$ ($\approx 420\,\mathrm{\mu mol\,mol^{-1}}$) and CH$_4$ ($\approx 1920\,\mathrm{nmol\,mol^{-1}}$) (NOAA Global Monitoring Laboratory data viewer)) and (2) the relatively large fraction of in situ $^{14}$C that forms CO in ice ($\approx 0.31$; Dyonisius et al., 2023; Hmiel et al., 2023). This makes $^{14}$CO the best species for investigating the in situ cosmogenic component of $^{14}$C in ice.

In situ $^{14}$CO in glacial ice is present at very low concentrations (a few molecules per gram of ice is typical; see Figs. 2 and 3), making measurements very analytically challenging. Prior studies have either worked with relatively small (a few kg) ice samples available from a single shared ice core (e.g., van der Kemp et al., 2002), resulting in relatively large uncertainties, or required dedicated ice coring campaigns to obtain large ice amounts (100 kg or more) from multiple parallel ice cores for high-precision measurements (e.g., Dyonisius et al., 2023). Dry extraction of air from ice has been used for smaller ice samples (van der Kemp et al., 2002), and melt-extraction has been used for large samples (Dyonisius et al., 2023). CO in the extracted air is separated, it is combusted to CO$_2$, this CO$_2$ is subsequently converted to graphite, and then the $^{14}$C/$^{13}$C or $^{14}$C/$^{12}$C ratio is measured via accelerator mass spectrometry. A detailed description of the ice core

$^{14}$CO measurement methodology can be found in Dyonisius et al. (2023).

## 2.2 Production of $^{14}$CO in glacial ice

Prior studies (Dyonisius et al., 2023; Hmiel et al., 2023) have presented detailed parameterizations of in situ cosmogenic $^{14}$CO production rates in glacial ice. This work uses the same parameterizations, which are described again here for the reader's convenience. The $^{14}$C production rate in ice via the neutron mechanism declines exponentially with depth, with the $^{14}$CO production rate calculated following Hmiel et al. (2023) as

$$P_n^{\mathrm{CO}}(h) = \Omega^{\mathrm{CO}} \cdot F_n \cdot S_n \cdot Q_C \cdot P_{n,\mathrm{SLHL}}^{\mathrm{Qtz}}(0) \cdot e^{-h/\Lambda_n}. \quad (1)$$

In this equation, $h$ is the mass-depth (in $\mathrm{g\,cm^{-2}}$), $\Omega^{\mathrm{CO}}$ is the fraction of total in situ $^{14}$C that forms CO (we use 0.31, following Hmiel et al., 2023) and $F_n$ is an adjustable dimensionless parameter that allows for tuning the neutron mechanism production rate within uncertainties (0.9–1.1 range). $S_n$ is the site-specific dimensionless scaling factor which describes the ratio of $^{14}$C production rate at the site of interest to $^{14}$C production rate at a sea-level high-latitude reference site; $S_n$ is determined using the model of Lifton et al. (2014). $Q_c$ TS3 is a factor that translates $^{14}$C production rate from quartz to ice using the difference in oxygen atom density (atoms $\mathrm{g^{-1}}$ TS4) between ice and quartz ($Q_c = 1.667$). $P_{n,\mathrm{SLHL}}^{\mathrm{Qtz}}(0)$ is the reference $^{14}$C production rate at the surface via the neutron mechanism in quartz at a sea-level high-latitude site; we use a value of $12.76\,\mathrm{atoms\,g\,Qtz^{-1}\,yr^{-1}}$ from the CRONUS-Earth project, which is defined for the 2001–2010 mean solar modulation and geomagnetic field conditions (Borchers et al., 2016). $\Lambda_n$ is the absorption mean free path of neutrons in

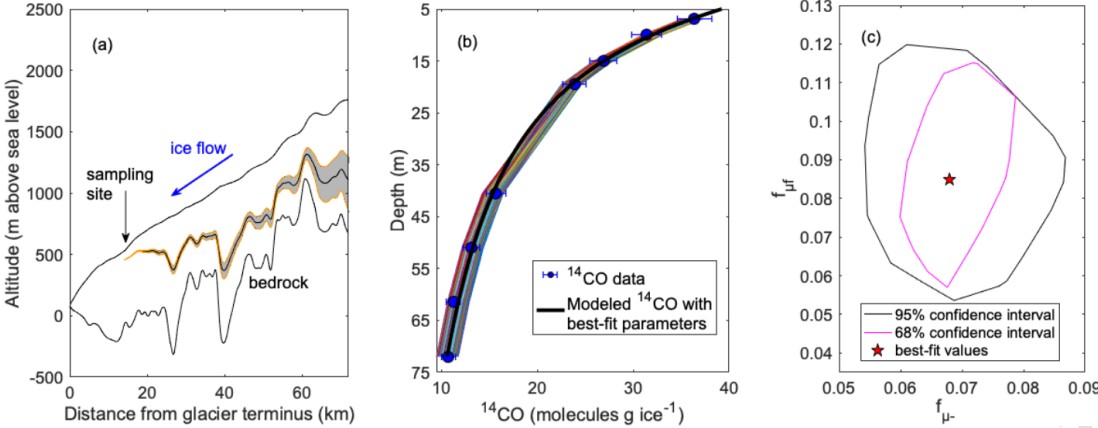

**Figure 2.** Overview of $^{14}$CO results from Taylor Glacier. **(a)** Ice parcel back-trajectories for the deepest (72 m) Taylor Glacier $^{14}$CO sample. The solid black line shows the best-estimate flow trajectory, and the shaded envelope represents the 68 % confidence interval (CI). **(b)** Comparison of Taylor Glacier $^{14}$CO measurements with model predictions for accepted scenarios. **(c)** Accepted ranges of $f_{\mu-}$ and $f_{\mu f}$. Figures modified from Dyonisius et al. (2023).

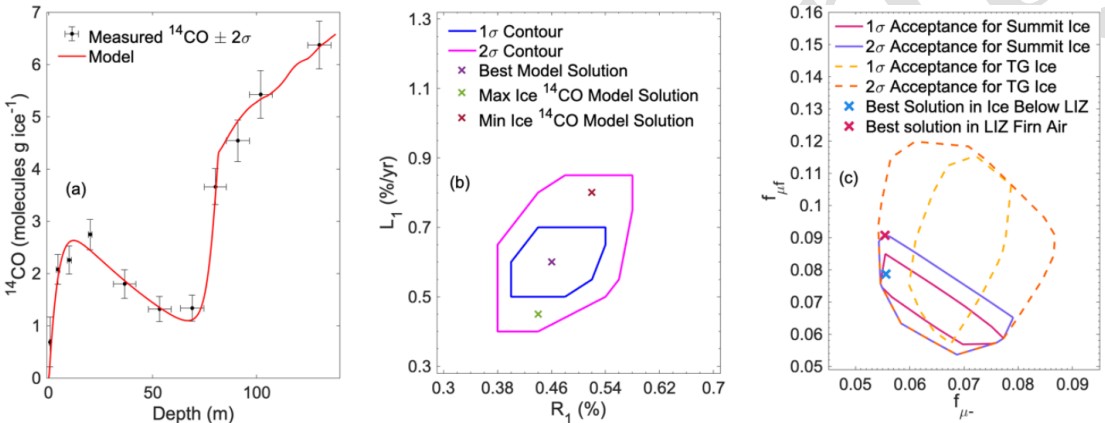

**Figure 3.** Overview of $^{14}$CO results from Summit Camp, Greenland CE1. **(a)** Measured $^{14}$CO content in ice grains and closed porosity along with a model fit. Horizontal error bars represent the depth range of firn and ice included in each sample. **(b)** Contour plot of the accepted ranges of the $R_1$ (initial retention) and $L_1$ (slow leakage) parameters in firn, together with the best-fit solution as well as solutions that result in maximum and minimum $^{14}$CO content in ice below the firn zone. **(c)** Contour plot of accepted ranges of $f_{\mu-}$ and $f_{\mu f}$ from Taylor Glacier (dashed lines) and after further constraints from Summit Camp measurements (solid lines).

ice; we use a value of $150\,\text{g cm}^{-2}$ (Lal et al., 1987; van de Wal et al., 2007).

For $^{14}$CO production by the muon mechanisms, we use a model developed by Balco et al. (2008) ("Balco model"), which incorporates parameterizations of Heisinger et al. (2002a, b). The $^{14}$C production rate via negative muon capture in these parameterizations is calculated using

$$P_{\mu-}(h) = R_{\mu-}(h) \cdot f_C\cdot, f_D \cdot f^* \tag{2}$$

where $R_{\mu-}(h)$ is the stopping rate of negative muons (muons $\text{g}^{-1}\,\text{yr}^{-1}$) at mass-depth $h$, $f_C$ is the chemical compound factor representing the probability that the stopped muon is captured by one of the target atoms, $f_D$ is the probability that the negative muon does not decay in the K shell before nuclear capture and $f^*$ is the effective probability for

production of the cosmogenic nuclide of interest after $\mu^-$ capture by the target nucleus. For production of $^{14}$C from $^{16}$O in ice, $f_C = 1$, $f_D = 0.1828$ and $f^* = 0.137$ (Heisinger et al., 2002a).

The $^{14}$C production rate via the fast muon mechanism is given by Heisinger et al. (2002b) as

$$P_{\mu f}(h) = \sigma_0 \cdot \beta(h) \cdot \phi(h) \cdot \overline{E}(h)^\alpha \cdot N, \tag{3}$$

where $\sigma_0$ is the reference nuclear reaction cross section at a muon energy of $1\,\text{GeV}$ (cm$^2$), $\phi(h)$ is the total muon flux at mass-depth $h$ (muons $\text{cm}^{-2}\,\text{yr}^{-1}\,\text{sr}^{-1}$), $\overline{E}(h)$ is the mean muon energy at mass-depth $h$ (GeV), $\alpha$ is a power factor that describes the energy dependence of the cross section (we use $\alpha = 0.75$, consistent with Dyonisius et al., 2023, and Heisinger et al., 2002b) and $N$ is the "number of target nuclei

per gram target mineral" (Hmiel et al., 2023). We use $\sigma_0 = 8.8\,\mu b = 8.8 \times 10^{-30}\,\text{cm}^2$ (Heisinger et al., 2002b) and $N = (6.022 \times 10^{23}\,\text{atoms mol}^{-1})/(18.02\,\text{g mol}^{-1})$ for ice. $\beta(h)$ is a unitless depth dependence factor ($\approx 0.9$ at our depths of interest, with only a slight dependence on depth), given by Heisinger et al. (2002b) as

$$\beta(h) = \frac{\overline{E(h)^{\alpha}}}{\overline{E(h)}^{\alpha}}. \tag{4}$$

Heisinger et al. (2002b) also provide an approximate function for $\beta(h)$, which is used in the Balco model and hence in our model:

$$\beta(h) = 0.846 - 0.015 \ln(h+1) + 0.003139(\ln(h+1))^2. \tag{5}$$

The Balco model incorporates Eqs. (2) and (3) and also provides the muon fluxes and energies as a function of mass-depth $h$ for a given site, using site atmospheric pressure as input.

## 2.3 Constraints on in situ $^{14}$CO production rates from measurements at Taylor Glacier, Antarctica

Recent studies at Taylor Glacier, Antarctica (an ice ablation site that exposes ancient ice at the surface), have provided measurements of $^{14}$C in ice older than 50 ka (Dyonisius et al., 2023; Petrenko et al., 2016). In such ice, any $^{14}$C from the snow accumulation site (from $^{14}$C-containing atmospheric gases trapped in air bubbles or from in situ cosmogenic production) has decayed away ($^{14}$C half-life is 5700 years), and the only measurable $^{14}$C originates from relatively slow in situ cosmogenic production by muons as the glacier transports the ice at large depths and somewhat faster production as the ice gradually rises toward the surface via ablation. Due to the relatively fast ice ablation rate of $\approx 20\,\text{cm yr}^{-1}$, the $^{14}$C contribution from the neutron production mechanism is negligible for ice deeper than 6 m. This presented an opportunity to use $^{14}$CO measurements in Taylor Glacier to constrain the muogenic $^{14}$CO production rates in ice in a natural setting.

Dyonisius et al. (2023) presented measurements of $^{14}$CO in Taylor Glacier ice between the surface and 72 m depth. An ice flowline model for Taylor Glacier (Buizert et al., 2012b) was used to reconstruct the possible range of trajectories for the sampled ice parcels (Fig. 2a). The Balco model was used to calculate $^{14}$CO production via the muon mechanisms as ice parcels followed the trajectories. As prior work suggested that muogenic $^{14}$C production rates from Heisinger et al. (2002a, b) may be too high when applied to ice (Petrenko et al., 2016), Dyonisius et al. (2023) introduced production rate adjustment factors $f_{\mu-}$ and $f_{\mu f}$ into production rate equations as follows:

$$P_{\mu-}^{\text{CO}}(h) = f_{\mu-} \cdot P_{\mu-}^{\text{Balco}}(h, P), \tag{6}$$

$$P_{\mu f}^{\text{CO}}(h) = f_{\mu f} \cdot P_{\mu f}^{\text{Balco}}(h, P). \tag{7}$$

Here $P_{\mu}^{\text{Balco}}(h, P)$ is the total $^{14}$C production rate (in atoms g$^{-1}$ yr$^{-1}$) in the Balco model for the respective muon mechanism at mass-depth $h$ and surface pressure $P$. $f_{\mu-}$ and $f_{\mu f}$ account for (1) the fraction of total $^{14}$C that forms $^{14}$CO ($\Omega^{\text{CO}}$) and (2) the adjustment factor for total $^{14}$C production rate.

To define the best-estimate $^{14}$CO production rate adjustment factors $f_{\mu-}$ and $f_{\mu f}$, Dyonisius et al. (2023) used a grid search approach, as follows. Using the best-estimate ice parcel back-trajectory (Fig. 2a), an expected $^{14}$CO depth profile was calculated for each combination of $f_{\mu-}$ and $f_{\mu f}$ between 0 and 0.2 at 0.001 resolution. The model results were then compared to $^{14}$CO measurements (Fig. 2b) with mean depths of 6.85 m or deeper (to avoid significant effects from the neutron mechanism), and a $\chi^2$ metric was used to determine the goodness of fit. To define the possible range of $f_{\mu-}$ and $f_{\mu f}$, Dyonisius et al. (2023) used a Monte Carlo approach, as follows. First, 10 000 possible ice back-trajectories were generated by perturbing ablation rates along the glacier according to their uncertainties (Fig. 2a). Next, a wide prior distribution for $f_{\mu-}$ and $f_{\mu f}$ was defined by starting with the best-estimate values and assuming a large and normally distributed 200 % uncertainty in these values. Thus, 100 000 Monte Carlo iterations of the model were then run, with each iteration randomly selecting a back-trajectory scenario and a pair of $f_{\mu-}$ and $f_{\mu f}$ from the prior distribution described above. All pairs of $f_{\mu-}$ and $f_{\mu f}$ that yielded $^{14}$CO depth profiles (Fig. 2b) that were within average measurement uncertainty ($1\sigma$ or $2\sigma$) from the best-fit solution were accepted (Fig. 2c).

## 2.4 Constraints on in situ $^{14}$CO retention and leakage in firn and production in ice at Summit Camp, Greenland

In situ cosmogenic $^{14}$C that is produced in the firn column above the lock-in depth can be lost to the atmosphere if it is able to leak out of the ice grains, resulting in low $^{14}$C retention into ice below the firn zone (e.g., de Jong et al., 2004; Petrenko et al., 2013, and references therein). Hmiel et al. (2023) used Summit Camp, Greenland to conduct the most comprehensive study to date of in situ cosmogenic $^{14}$C in the firn, with a focus on $^{14}$CO. This study measured $^{14}$CO in the ice grains in the firn matrix, in firn air, as well as in bubbly ice below the firn zone. Very large firn and ice samples (200–300 kg) were used for $^{14}$CO analysis, to provide sufficiently large numbers of $^{14}$C atoms for precise $^{14}$C measurements. Figure 3a shows the $^{14}$CO results for samples from the firn, firn-ice transition and bubbly ice below the firn zone. In the shallowest firn, $^{14}$CO increases rapidly with depth owing mainly to production by the neutron mechanism, reaching a peak in the 10–20 m depth range. Beyond 20 m, $^{14}$CO in the firn matrix declines gradually with depth in the diffusive part of the firn, reflecting leakage of in situ $^{14}$C from the ice grains. $^{14}$CO increases rapidly in the lock-in zone ($\approx 70$–

80 m), reflecting addition of $^{14}$CO from trapped air. Below the lock-in zone, $^{14}$CO in the ice continues to increase gradually due to deeper production by the muon mechanisms.

To interpret the Summit Camp $^{14}$CO results, Hmiel et al. (2023) employed a firn gas transport model that can also characterize trapped air in ice below the firn zone (Buizert et al., 2012a). Production of in situ $^{14}$C following the systematics described in Sect. 2.2 and tracking of $^{14}$C in ice grains and porosity was implemented in this model. With regard to $^{14}$C loss from ice grains in the firn, it was found that the model–data agreement was best if two separate loss processes were parameterized in the model: a fast process, with a timescale $< 1$ year and an additional slow process. This was described in the model using parameters $R_1$ and $L_1$. $R_1$ represents the fraction of in situ $^{14}$C in the ice grains that is initially retained. The fraction of in situ $^{14}$C in ice grains that is lost rapidly, given by $1 - R_1$, leaks out from the ice grains into the porosity at every model time step (0.5 year). $L_1$ represents the fraction of the initially retained $^{14}$C that is lost more slowly from the ice grains over the course of 1 year. Hmiel et al. (2023) used a grid search approach to constrain the possible ranges of $R_1$ and $L_1$ at Summit Camp (Fig. 3b), showing that $> 99\%$ of in situ $^{14}$C is lost rapidly from the ice grains, while the remaining $\approx 0.5\%$ ($R_1$) of in situ $^{14}$C continues to leak out slowly at a rate of $\approx 0.6\%\,\mathrm{yr}^{-1}$ ($L_1$). Hmiel et al. (2023) argued that the rapid loss is best explained by the process of gas diffusion through ice and suggested that the $\approx 0.5\%$ of $^{14}$C that is initially retained may be trapped in microbubbles or by impurities at dislocations or grain boundaries and is released via the process of recrystallization.

Summit Camp $^{14}$CO measurements in ice below the firn zone also provided an opportunity to test muon mechanism $^{14}$CO production rate estimates from Taylor Glacier. For Summit Camp ice samples, the contribution from trapped atmospheric $^{14}$CO is important ($\approx 25\%$–$40\%$ of total), and uncertainties in the atmospheric $^{14}$CO history interfere with precise constraints on $f_{\mu-}$ and $f_{\mu f}$. Nevertheless, by trialing the Taylor Glacier sets of accepted $f_{\mu-}$–$f_{\mu f}$ TS5 pairs in combination with several possible atmospheric $^{14}$CO histories, Hmiel et al. (2023) were able to further narrow the possible ranges of $f_{\mu-}$ and $f_{\mu f}$ (Fig. 3c).

# 3 In situ cosmogenic $^{14}$CO in ice cores as a possible proxy for GCR flux variability

## 3.1 Basic concept for using $^{14}$CO in ice cores as a GCR flux proxy

As the Summit Camp, $^{14}$CO results summarized above illustrate, the retention of in situ cosmogenic $^{14}$CO through the upper firn column is very low. This means that the majority of in situ $^{14}$CO found in ice below the firn zone originates from production by muons below the lock-in depth, where this $^{14}$CO can no longer escape to the atmosphere. If the firn layer is sufficiently thick ($\approx 90$–$100\,\mathrm{m}$ actual depth or $\approx 65\,\mathrm{m}$ ice equivalent depth), the muons penetrating below the firn must have an energy of $\approx 15\,\mathrm{GeV}$ or greater at the surface (e.g., Rogers and Tristam, 1984). Such muons originate from primary GCR particles with energies of $\approx 100\,\mathrm{GeV}$ or greater (Gaisser et al., 2016). The part of the GCR flux possessing such energies is not affected appreciably by either the geomagnetic or the heliospheric magnetic fields. In situ cosmogenic $^{14}$CO content in ice cores drilled at such sites thus can serve as a proxy of variations in the primary GCR flux. This proxy is in principle free of the confounding effects discussed in the Introduction for other past GCR flux indicators.

Several considerations are important for site selection in order to increase the likelihood of success with this proxy. First, the in situ $^{14}$CO signal must be maximized to help with measurement sensitivity as well as to reduce interference from the trapped atmospheric $^{14}$CO component. Second, the site must have a thick firn column. This is needed to ensure that $^{14}$CO below the firn zone is produced only by muons originating from primary GCRs that are sufficiently energetic to be unaffected by solar magnetic field variations. Third, there should be as little in situ $^{14}$CO retained from the shallow firn as possible. $^{14}$CO produced in the shallow firn originates from neutrons or lower-energy muons that are affected by solar magnetic field variations and may complicate interpretation. Fourth, ideally the site must be glaciologically stable over time in terms of accumulation rate and lock-in depth. Large temporal variations in these parameters may introduce additional uncertainties in the interpretation, as they affect the predicted in situ $^{14}$CO content.

Considering the above, ice dome sites in the east Antarctic interior are most promising for attempting to examine past GCR flux variability using $^{14}$CO in ice cores. Low accumulation rates at such sites maximize cosmogenic exposure times and thereby the in situ $^{14}$CO signal. These sites also tend to have sufficiently thick firn columns (e.g., Buizert, 2013). The combination of low accumulation rate and thick firn column results in very long ice layer transit times through the firn, maximizing the chance that in situ $^{14}$CO produced by neutrons and low-energy muons in the shallow firn would be lost. Finally, dome sites are free of complications of upstream ice advection and ice core water stable isotope records, suggest that the interior east Antarctic climate has been stable over the last few thousand years (recent decades excepted) (e.g., Jouzel et al., 2001).

## 3.2 Using model predictions to explore Dome C, Antarctica, as a test case for the $^{14}$CO GCR flux proxy

Dome C, Antarctica, is a site that meets the criteria needed for the $^{14}$CO GCR flux proxy to be viable. It has been glaciologically very well characterized as a result of previous ice coring campaigns (e.g., EPICA Community Members, 2004)

and has well-established logistical access owing to the permanent Concordia Station. Further, a scientific ice drilling expedition is planned for the near future to Dome C for the purpose of $^{14}$CO reconstruction at this site. We therefore use Dome C as an example site for more detailed model-based exploration of the $^{14}$CO past GCR flux proxy. We first applied the full firn-ice model mentioned above (Buizert et al., 2012a; Hmiel et al., 2023) to explore the (unwanted) contribution of $^{14}$CO originating from production in the shallow firn as well as trapped atmospheric $^{14}$CO to the overall $^{14}$CO signal in ice below the firn zone. In the model, we used an accumulation rate of 3.2 cm ice equivalent yr$^{-1}$ and the firn density profile from the FIRETRACC project (EU FIRETRACC Campaign participants, 2006), and we tuned the firn gas diffusivity profile based on a combination of available $CO_2$, $CH_4$, CFC-11, CFC-12, CFC-113, $CH_3CCl_3$, $SF_6$ and $\delta^{15}N$ of $N_2$ measurements (EU FIRETRACC Campaign participants, 2006). For parameters relevant to in situ $^{14}$CO, we used $F_n = 1.03$, $R_1 = 0.44\%$ and $L_1 = 0.45\%$ yr$^{-1}$ (see Sect. 2), which was the combination of values at Summit Camp that maximized the amount of in situ $^{14}$CO produced in the shallow firn that is retained into ice below the firn zone (Fig. 3b). For muogenic $^{14}$CO production, we used $f_{\mu-} = 0.065$ and $f_{\mu f} = 0.07$, which are mid-range choices from the possible range of values that were consistent with both Taylor Glacier and Summit Camp measurements (Fig. 3c). We used a constant concentration of 12 molecules cm$^{-3}$ (standard temperature and pressure CE2) for the atmospheric $^{14}$CO history, which is the average of the longest available Antarctic atmospheric $^{14}$CO record (Manning et al., 2005).

Figure 4a shows model-calculated $^{14}$CO content that represents the sum of $^{14}$CO in ice grains and closed porosity (this is what measurements done with a melt-extraction approach would provide). The solid black line shows results with both in situ and atmospheric $^{14}$CO included in the model. There is a sharp $^{14}$CO peak at $\approx 9$ m depth that represents $^{14}$CO in ice grains and is driven by intense $^{14}$CO production by the neutron mechanism in near-surface firn. $^{14}$CO then declines to near zero by $\approx 70$ m due to slow leakage out of ice grains (controlled by the $L_1$ parameter in the model). At depths $> 80$ m, the amount of closed porosity starts to increase, and this increases $^{14}$CO by trapping of $^{14}$CO from open porosity and by allowing more in situ $^{14}$CO to be retained. This process further accelerates at $\approx 95$ m, which is the lock-in depth at Dome C. Below the close-off depth at Dome C ($\approx 100$ m), $^{14}$CO content continues to increase due to production by muons, rising to 8.5 $^{14}$CO molecules g$^{-1}$ ice CE3 at the deepest modeled level (110 m).

The dashed blue line shows the expected contribution to total $^{14}$CO from in situ $^{14}$CO originating only from the shallower part of the firn. This was assessed by setting the atmospheric $^{14}$CO history to zero and setting in situ production rates to zero for depths $> 54$ m ice equivalent. This contribution is $< 0.5$ $^{14}$CO molecules g$^{-1}$ ice and is due to $^{14}$CO that leaks out from ice grains in the shallow firn, diffuses

into the deep firn and is subsequently trapped in air bubbles. The contribution from trapped atmospheric $^{14}$CO (dotted pink line; assessed by turning off in situ production in the model) is $< 1.2$ $^{14}$CO molecules g$^{-1}$ ice. $^{14}$CO originating from the sum of shallow in situ cosmogenic production and air trapping (solid red line) is $< 1.6$ $^{14}$CO molecules g$^{-1}$ ice at all depths below the firn zone.

We next examined the in situ cosmogenic $^{14}$CO component at Dome C arising from production by deep-penetrating muons, as well as its suitability for detecting changes in the past GCR flux. As this approach involved generating thousands of simulated data sets (see Sect. 3.3 below), we created a simple and computationally efficient ice-only model of in situ cosmogenic $^{14}$CO for this test of the proxy concept. This ice-only model has its starting (shallowest) depth in the lock-in zone and assumes an initial $^{14}$CO content of zero. $^{14}$CO production in the model is implemented following parameterizations described above in Sect. 2, with production rates within the range constrained by Taylor Glacier and Summit Camp results. The model assumes that all of the in situ $^{14}$CO is retained and also includes $^{14}$C radioactive decay. The model defines annual ice layers and shifts these layers downward on an annual basis following the ice layer age scale for Dome C from Buizert et al. (2018). For the purposes of this test of the proxy concept, we set the deepest model depth at 300 m, as this is the practical limit for light ice coring projects that do not use drilling fluid and the deepest depth in the planned fieldwork. The exact starting depth of the model was chosen by comparing predictions of this ice-only model with predictions from the full firn-ice model in the 100–110 m depth range (below close-off depth) when using the same muogenic $^{14}$CO production rates and setting atmospheric $^{14}$CO history to zero; using 96.5 m for the starting depth yields the best match.

Figure 4b illustrates predictions of the simple ice model for a few scenarios involving different combinations of $f_{\mu-}$ and $f_{\mu f}$ as well as different production rate histories (representing past GCR flux variations), and Fig. 4c illustrates the time-variable production rate scenarios reflected in Fig. 4b. Because in situ cosmogenic $^{14}$CO production takes place at the full range of modeled depths (with production rate declining with depth as illustrated in Fig. 1b), the $^{14}$CO values at each depth represent a time integral of production rate minus the $^{14}$C decay rate. As expected, $^{14}$CO content increases most rapidly at the shallowest depths, followed by a broad peak in the 200–250 m depth range. For deeper ice, the rate of $^{14}$CO removal via radioactive decay exceeds the rate of production by muons, and $^{14}$CO values gradually decline. The modeled ice layers at Dome C span an age range of 7283 years, meaning that an ice core $^{14}$CO record reaching 300 m depth could offer information about past GCR flux variations for most of the Holocene.

Predicted $^{14}$CO content originating from deep-penetrating muons is between 20 and 30 molecules g ice$^{-1}$ for most of the modeled depth range. This means that the $^{14}$CO con-

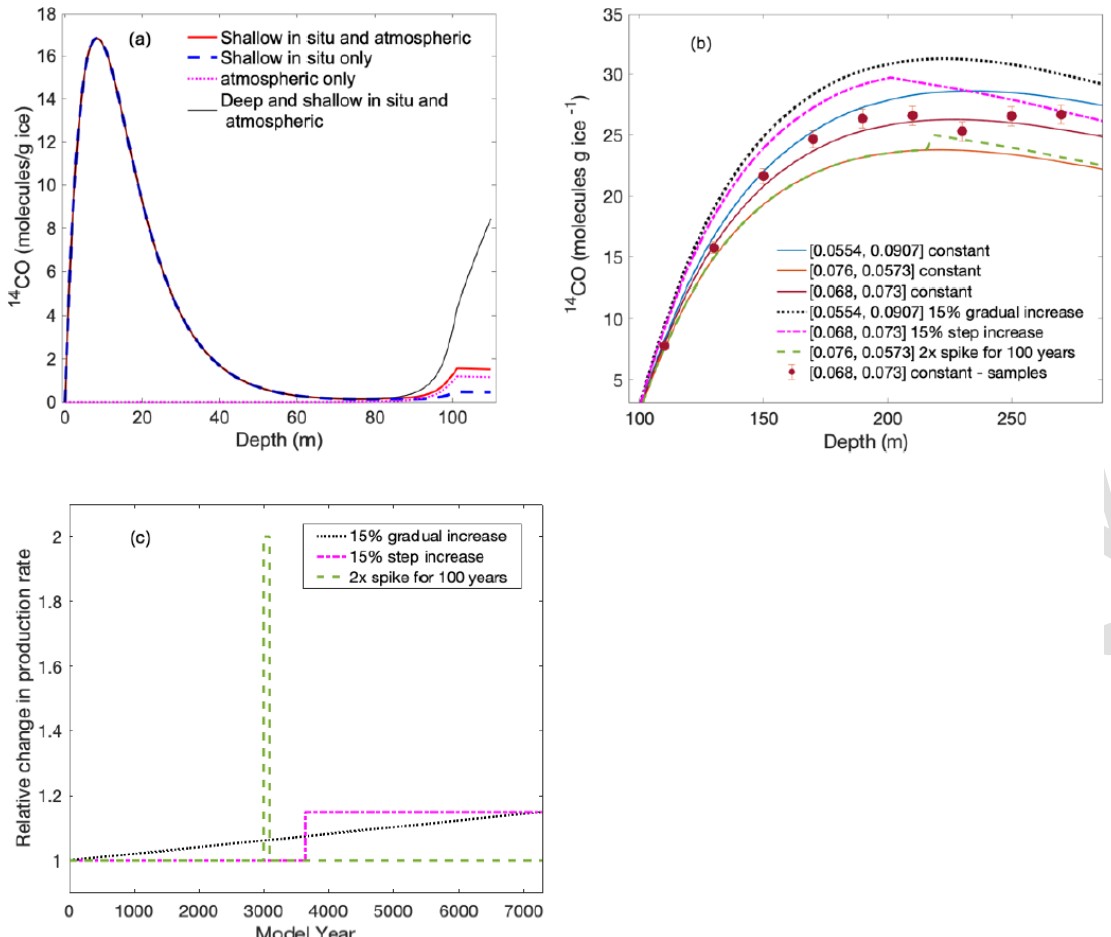

**Figure 4.** Predicted $^{14}$CO content at Dome C. **(a)** Predicted $^{14}$CO content in ice grains + closed porosity (bubbles) from the full firn-ice model considering all $^{14}$CO contributions as well as contributions from individual $^{14}$CO components. **(b)** Predicted in situ $^{14}$CO content in ice below the firn zone from a simple ice-only model. The legend indicates the combination of $[f_{\mu-}, f_{\mu f}]$ values (see Sect. 2) used in each model run, as well as whether the production rate was assumed to be constant (solid lines) or variable (dashed and dotted lines) in time. Markers illustrate what sample measurements might look like assuming 20 m depth averaging and a random $1\sigma$ measurement error of 3 %. **(c)** Time-variable production rate scenarios used to generate the corresponding depth–$^{14}$CO profiles in panel **(b)**.

tribution arising from trapped atmospheric $^{14}$CO and $^{14}$CO production in shallow firn ($< 1.6$ molecules g ice$^{-1}$; Fig. 4a) would contribute $< 8$ % of total $^{14}$CO and is unlikely to interfere with $^{14}$CO signal interpretation. We note that there were three prior $^{14}$CO measurements that were made on the Dome C ice core in the depth range we are considering (de Jong et al., 2004). Those measurements were made on much smaller (1–2 kg) samples than Taylor Glacier and Summit measurements and thus were associated with very large uncertainties (50 %–100 %, considering only uncertainty reported for $^{14}$C activities). That said, de Jong et al. (2004) reported $^{14}$CO values in the 15–30 molecules g ice$^{-1}$ range for these three samples and concluded that there was no detectable in situ $^{14}$C retention from the firn, consistent with our model predictions.

Figure 4b illustrates that the absolute $^{14}$CO content in the ice as well as the depth of the $^{14}$CO peak depend on the balance of production rates from the negative muon capture and fast muon mechanisms (solid lines); this is controlled by the $f_{\mu-}$ and $f_{\mu f}$ parameters in the model. In the modeled depth range, the fast muon mechanism is relatively more important (Fig. 1b), so maximizing $f_{\mu f}$ at the expense of $f_{\mu-}$ (within the range in Fig. 3c) increases total $^{14}$CO and shifts the peak slightly deeper. Despite these differences, the shape of the depth–$^{14}$CO curves remains largely similar. Figure 4b also illustrates a few scenarios where the production rates (controlled by GCR flux) vary in time (dashed and dotted lines). For time-variable production rates for the purpose of this illustration (Fig. 4c), we trialed (1) a scenario where production by each muon mechanism increases at a linear rate over the entire duration of the model run, reaching 15 % higher rates by the end of the run (black dotted line), (2) a scenario where there is a 15 % step increase in production rates halfway through the model run (pink dash-dot

line) and (3) a scenario where there is a 2 times transient step increase in production rates that takes places between 3000 and 3100 years in the model run (dashed green line). As can be seen, all of these types of variations produce depth–$^{14}$CO profiles that are distinct in their shape from the constant production rate scenarios.

### 3.3 Analysis of sensitivity of ice core $^{14}$CO measurements at Dome C to variations in past GCR flux

We compare several time-varying scenarios to the baseline model of a constant GCR flux with muonic production rates ($f_{\mu-}$, $f_{\mu f}$) that are consistent with ice core $^{14}$CO measurements in both Dyonisius et al. (2023) and Hmiel et al. (2023). As shown in Fig. 4b, deviations from the baseline model can be produced by temporal variations in the GCR flux. However, in the presence of a steady-state flux, uncertainties in the muonic production rates also create deviations from the baseline model. While the normalization of the depth–$^{14}$CO profile is affected by both the temporal variations in the GCR flux and the production rates, the shape of the profile is more sensitive to temporal variations in the flux. Therefore, we develop an analysis that is sensitive to the shape of the $^{14}$CO profile as a function of depth.

To discriminate the steady-state GCR scenario $H_0$ from the time-varying scenario $H_1$, we construct a test statistic using a Bayes factor (Jeffreys, 1998; Kass and Raftery, 1995):

$$B_{01} = \frac{P(c^{\text{TS6}}|H_0)}{P(c|H_1)} = \frac{\int d\theta_0\, P(c|\theta_0, H_0)\, P(\theta_0|H_0)}{\int d\theta_1\, P(c|\theta_1, H_1)\, P(\theta_1|H_1)}. \quad (8)$$

In this expression, $c = \{c(h_j)\}$ is a $^{14}$CO profile measured as a function of discrete depths $h_j = \{h_1, \ldots, h_N\}$. The Bayes factor computes the ratio of the marginal probabilities of measuring $^{14}$CO profile $c$ given the steady-state and time-varying scenarios $H_0$ and $H_1$. If the data provide greater evidence for the steady-state hypothesis $H_0$, $B_{01} > 1$, and if the time-varying hypothesis is supported, $B_{01} < 1$.

During the calculation of $B_{01}$, each marginal probability, $P(c|H_i)$, can be factorized into two terms: a conditional probability $P(c|\theta_i, H_i)$, where $\theta_i$ lists the free parameters of GCR flux model $H_i$, and a "prior" probability distribution $P(\theta_i|H_i)$. To complete the calculation, we integrate over the possible values of the parameters $\theta_i$ in each model $H_i$. The prior probabilities specify the allowed ranges of parameters $\theta_i$ in model $H_i$ and allow us to weight the calculation toward more probable values of the model parameters. Note that we are free to choose the functional form of the prior distributions using theoretical considerations, past measurements or even our subjective degree of belief in the most likely values of the parameters of a given model. In this work, we use non-informative (or flat) prior distributions that do not favor any particular values of the model parameters, beyond restricting their ranges to physically motivated regions.

In the sensitivity calculation, the muonic production rates ($f_{\mu-}$, $f_{\mu f}$) are nuisance parameters folded into both $\theta_0$ and $\theta_1$. Using the confidence intervals on ($f_{\mu-}$, $f_{\mu f}$) from Dyonisius et al. (2023) and Hmiel et al. (2023), we can factorize the prior probability $P(\theta_i|H_i)$ for model $i$ into a joint prior $P(f_{\mu-} f_{\mu f}|H_i)$ and a set of independent priors dependent on the parameters of the model. For example, the joint prior $P(f_{\mu-} f_{\mu f}|H_i)$ is given by the "$2\sigma$ acceptance from Summit ice" contour in Fig. 3c. If we wish to test a cosmic ray model $H_1$ with a flux that varies linearly in time, the model includes an additional free parameter $a$ representing the rate of change of the flux as a function of time. In the calculation of the Bayes factor, the prior distribution of $a$ is a uniform probability density function:

$$P(a|H_1) = \begin{cases} \frac{1}{a_{\max}-a_{\min}}, & a \in [a_{\min}, a_{\max}]; \\ 0, & \text{otherwise.} \end{cases} \quad (9)$$

Here $a_{\min}$ and $a_{\max}$ represent the allowed range of values we consider for the rate of change of the flux. We use a uniform distribution for $P(a|H_1)$ because it is unbiased, giving equal weight to all values of $a$ between $a_{\min}$ and $a_{\max}$.

Our calculation assumes the ice core $^{14}$CO measurements are depth-averaged over 20 m, and each measurement has independent Gaussian uncertainties with relative sizes of 2 % at $1\sigma$. The 20 m depth averaging is assumed because this would provide the needed amount of ice for high-precision $^{14}$CO measurements ($\approx 140$ kg with a 10 cm diameter ice core). Recent improvements in analytical techniques for ice core and atmospheric $^{14}$CO measurements (Petrenko et al., 2023, 2021) make 2 % uncertainties achievable, although for completeness we also repeat the calculations assuming 3 % $1\sigma$ uncertainties. The conditional probability of observing a $^{14}$CO profile $c$ given GCR model $H_i$ with parameters $\theta_i$ is

$$P(c|\theta_i, H_i) =$$
$$\prod_{j=1}^{N} \frac{1}{\sqrt{2\pi}\,\sigma_j} \exp\left\{ -\frac{1}{2}\left( \frac{c_j - \hat{c}(h_j|\theta_i, H_i)}{\sigma_j} \right)^2 \right\}. \quad (10)$$

Here $\hat{c}(h_j|\theta_i H_i)$ is the expected $^{14}$CO profile at depth $h_j$, which we compute using the model, while $c_j = c(h_j)$ is the observed $^{14}$CO concentration at depth $h_j$. Since the measurement uncertainties are independent and Gaussian, the probability is the product of $N$ independent Gaussian probability density functions over the $N$ measurements in the depth profile. Multiplying this probability by the prior distributions of the nuisance parameters ($f_{\mu-}$, $f_{\mu f}$) and the allowed prior ranges of the model parameters (such as the slope of the linear change in the GCR flux) allows us to account for both systematic and statistical uncertainties in the measurement.

We calculate our sensitivity to a given GCR scenario as follows:

1. We produce $5 \times 10^6$ random realizations of the $^{14}$CO profile at Dome C, assuming a constant production

**Table 1.** Simulated sensitivity to temporal changes in the GCR flux. We report the magnitude of GCR flux changes in time-varying models required to produce a $3\sigma$ or $5\sigma$ detection in at least 50 % of simulated data sets, assuming 2 % (3 %) relative uncertainties in the $^{14}$CO measurements. For example, to produce a $3\sigma$ detection of a linearly increasing or decreasing GCR flux, the rate of change of the flux must be at least 4 % (5 %) over 7 ka.

| Difference from baseline model | Sensitivity | |
|---|---|---|
| | $3\sigma$ (> 50 % of trials) | $5\sigma$ (> 50 % of trials) |
| Linear increase over 7 ka | 4 % (5 %) | 6 % (7 %) |
| Step-like increase at 3.5 ka | 4 % (5 %) | 6 % (7 %) |
| Impulsive increase lasting 100 years at 3.5 ka | 250 % (350 %) | 350 % (460 %) |

rate but accounting for the systematic uncertainties in ($f_{\mu-}$, $f_{\mu f}$). The profiles are generated with depth averaging of 20 m, and relative measurement uncertainties of $\sigma_j/c_j = 2$ % and 3 % are both investigated.

2. For each time-varying model under consideration, we compute a distribution of Bayes factors $B_{01}$ using the random constant-flux data sets. This provides us with a distribution of the Bayes factor when the null constant-flux hypothesis $H_0$ is true.

3. We next produce a large set of independent $^{14}$CO profiles assuming the alternative time-varying hypothesis $H_1$ is true and compute the Bayes factor $B_{01}^*$ for each simulated data set. We expect that $B_{01}^*$ will be much smaller than $B_{01}$, on average, since $P(c|H_1) > P(c|H_0)$ when the alternative hypothesis $H_1$ is true.

4. For each $B_{01}^*$, we compute the tail probability, or $p$ value, that gives the probability that a constant flux model will produce a Bayes factor smaller than the time-varying model purely by a chance statistical fluctuation, i.e.,

$$p = P\left(B_{01} < B_{01}^* | H_0\right). \tag{11}$$

The reported sensitivity of a given model is the value of the model parameter(s) in which at least 50 % of simulated data sets yield $p < 10^{-3}$ ($3\sigma$ evidence against the steady-state model). We also report the value of the model parameter(s) yielding $p < 3 \times 10^{-7}$, corresponding to a $5\sigma$ discovery of a time-varying flux. This "calibration" of the Bayes factor accounts for the chance probability that a steady-state flux could produce a false positive report of a time-varying flux.

We investigated scenarios involving (1) a linear GCR flux increase over the entire duration of the record, (2) a step-like increase at approximately the mid-point of the record and (3) a brief (100-year) burst in the GCR flux. The results are reported in Table 1. For a scenario $H_1$ where the GCR flux increases linearly with time and assuming 2 % (3 %) relative uncertainties in the measured $^{14}$CO profile, a flux increase $a = 4$ % (5 %) over 7 ka is required to produce a $3\sigma$ evidence of a non-steady flux in at least 50 % of simulated data sets.

For a $5\sigma$ detection, the rate of change of the flux must be at least $a = 6$ % (7 %). We also investigated and found similar sensitivities for a scenario involving a step-like increase in the GCR flux at 3.5 ka. Much larger GCR flux changes are required for detection in the impulsive burst scenario: 250 % (350 %) for $3\sigma$ evidence. This is likely due to the large amount of temporal averaging ($\approx 700$ years) that is imposed by the 20 m depth averaging for the measurements and the fact that the $^{14}$CO content at each depth level represents a time integral of production rates. We further note that improving the relative uncertainty in the $^{14}$CO measurement from 3 % to 2 % has a minor effect on the sensitivity to linear and step-like increases in the GCR flux, but the change in sensitivity to burst-like increases in the flux is substantial.

## 4 Conclusions

$^{14}$CO in ice cores at low-accumulation sites such as Dome C, Antarctica, has a good potential to provide a test of the assumption of GCR flux constancy over the Holocene and to serve as a proxy for past variations in the GCR flux on timescales of a few thousand years. $^{14}$CO measurements in the proposed approach would be most sensitive to gradual linear or step-like changes in the GCR flux, in principle allowing us to test the assumption of GCR flux constancy to 5 % or better. This would represent a large improvement over the $\approx 30$ % uncertainty associated with constraints from meteorite measurements. Because our approach involves a large amount of temporal averaging, sensitivity to short-lived GCR bursts is much worse. However, such bursts (if present) would have been captured by high-resolution records of other cosmogenic nuclides such as ice core $^{10}$Be and tree-ring $^{14}$C.

We note that cosmogenic nuclides produced in the atmosphere such as $^{10}$Be are primarily sensitive to the GCR flux below 10 GeV, while the $^{14}$CO proxy discussed here is sensitive to the flux above 100 GeV. The extent to which temporal variations in the GCR flux above 100 GeV would produce proportional changes below 10 GeV, while beyond the scope of this paper, is an interesting question to consider, as the answer depends on the origin of the temporal variations. Since the diffusion length of cosmic rays increases with energy, it is reasonable to expect that a constant GCR flux at high energy

is likely to imply a constant GCR flux below 10 GeV, while a time-varying flux above 100 GeV could still be consistent with a constant flux at or below 10 GeV.

For the most precise results, the $^{14}$CO proxy approach requires an ice dome site that is glaciologically stable (accumulation rate, lock-in depth) over the duration of the GCR flux reconstruction. Although our work indicates that the $^{14}$CO GCR flux proxy is likely to provide useful results for most of the Holocene, we expect that GCR flux reconstructions beyond the Holocene with this approach would be more challenging, owing to (1) the need for drilling fluid to obtain ice below $\approx 300$ m, which would greatly increase logistical requirements and introduce added challenges of CO contamination from the drilling fluid; (2) glaciological changes beyond the Holocene; and (3) reduced $^{14}$CO signal at greater depths due to $^{14}$C radioactive decay.

*Code and data availability.* Code for the firn and ice models as well as for the statistical analysis used in this study is available from https://github.com/14CO/Dome-C-Sensitivity TS7.

The simulated data sets created as part of the statistical analysis in this study are available from https://github.com/14CO/Dome-C-Sensitivity.

*Author contributions.* VVP and SB developed the $^{14}$CO GCR flux proxy concept. VVP wrote the code for the simple ice model and performed firn and ice model simulations. SB developed the approach, wrote the code for and performed statistical analyses. CB provided firn model tuning and ice layer age scale for Dome C. VVP and SB wrote the paper, with input from all other authors.

*Competing interests.* The contact author has declared that none of the authors has any competing interests.

ther geographical representation in this paper. While Copernicus Publications makes every effort to include appropriate place names, the final responsibility lies with the authors.

*Special issue statement.* This article is part of the special issue "Ice core science at the three poles (CP/TC inter-journal SI)". It is a result of the IPICS 3rd Open Science Conference, Crans-Montana, Switzerland, 2–7 October 2022.
OR
This article is part of the special issue "Ice core science at the three poles (CP/TC inter-journal SI)". It is not associated with a conference. TS8

*Acknowledgements.* This work was funded by the University of Rochester Bridging Fellowship (to Vasilii V. Petrenko) and US NSF Award OPP-2146131 (to Vasilii V. Petrenko and Segev BenZvi). We thank I. Usoskin, R. Muscheler, G. Balco, J. Stone and N. Lifton TS9 for helpful discussions.

*Financial support.* This research has been supported by the University of Rochester (Bridging Fellowship to Vasilii V. Petrenko) and the National Science Foundation (grant no. OPP-2146131).

*Review statement.* This paper was edited by Hubertus Fischer and reviewed by Ilya Usoskin and one anonymous referee.

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

## Remarks from the language copy-editor

## Remarks from the typesetter

**TS7** Please clarify whether the data set/code is your own. If yes, please provide a DOI in addition to your GitHub URL since our reference standard includes DOIs rather than URLs. If you have not yet created a DOI for your data set/code, please issue a Zenodo DOI (https://help.github.com/en/github/creating-cloning-and-archiving-repositories/referencing-and-citing-content). If the data set/code is not your own, please inform us accordingly. In any case, please ensure that you include a reference list entry corresponding to the data set including creators, title, and date of last access.