# Peer review of "The potential of *in situ* cosmogenic 14CO in ice cores as a proxy for galactic cosmic ray flux variations"

_EGUsphere, 2023_

## Author Comment (AC2)

Referee comments in regular font
*Our responses in italics*

**RC2**: 'Comment on egusphere-2023-3126', Ilya Usoskin, 06 Feb 2024
The manuscript by Petrenko et al. presents a programmatic paper presenting and quantifying a novel approach to measuring 14CO in ice as a proxy of the long-term flux of high-energy (>100 GeV) cosmic rays. It is shown that this method is feasible and can indeed be used to study the GCR consistency on the time scale of thousands of years. The authors describe the physics behind the approach very well and provide a quantitative assessment of the method's sensitivity to demonstrate that it is sufficient for the task. The authors also correctly specify the related challenges.
I found this work very important expectedly becoming a reference paper for the new method. I am happy to recommend this manuscript for potential acceptance subject to a minor revision related to some clarifications in the methodological description as specified below.
*We thank Dr. Usoskin for his detailed and helpful review and address individual comments below*

A reader would benefit from a brief general description of how 14CO is measured in ice.
*We will add a brief description in the revised manuscript*

Line 24: "insensitive" -- > "almost insensitive".
*We will make this change in the revised manuscript*

Line 33: after "solar irradiance" a reference to Wu et al. (2018b, doi: 10.1051/0004-6361/201832956) can be added.
*We will add this reference*

Line 35: please add a reference to a review by Usoskin (2023, doi: 10.1007/s41116-023-00036-z) focused on the cosmogenic method for solar activity reconstructions.
*We will add this reference*

Line 44: In addition to meteoritic studies, cosmogenic isotopes in lunar rocks can provide an estimate of the very long-term (mega-years) flux of cosmic rays (see, e.g., Poluianov et al., 2018, doi: 10.1051/0004-6361/201833561 and references therein). This method is free of geomagnetic shielding and uncertainties related to the orbit and erosion, but of course, is strongly affected by solar modulation. This can be briefly mentioned here in addition to the meteoritic data.
*We will add a brief mention of this study to the discussion*

Line 57: the statement about the isotropy of the GCR flux at the level of 1 permil needs clarification and a reference. The flux of GCR (in the GeV energy range) near Earth has a level of anisotropy of about 1% due to the orbital motion and diffusion+convection of particles by solar wind. Probably, the authors' statement is related to higher energies. A reference is needed.
*The reviewer is correct that we are referring to the GCR flux at 100 GeV and beyond. There are two anisotropic signals observable at these energies:*

1. *A dipolar anisotropy due to the Earth's orbit around the Sun, producing an annual modulation due to Earth's relative motion through the "wind" of the local GCR flux.*

2. *An approximately dipolar anisotropy (with statistically significant components on angular scales down to a few degrees), most likely due to cosmic-ray diffusion from unresolved sources of cosmic rays within roughly 1 kpc of the solar system. The amplitude of the dipole, projected onto the celestial equator, varies between $10^{-4}$ and $10^{-3}$ between several hundred GeV and 1 PeV. Above 1 PeV, to 10 EeV (the highest reported energies) the dipole increases to $10^{-2}$.*

*A technical detail is that the solar anisotropy is a contaminant of the galactic anisotropy, but it can be estimated and removed by transforming cosmic-ray arrival directions to the so-called "anti-sidereal" reference frame (details are available in Farley & Storey, Proc. Phys. Soc. 67:996, 1954, Guillian+ Phys. Rev. D75:062003, 2007, and Aartsen+ ApJ 826:220, 2016).*

*We will add relevant references in the text that document the 0.1% GCR anisotropy at TeV energies and clarify the energy range this applies for.*

Line 113: "Hmiel et al. (2023)" -- > "Hmiel et al., 2023)".
*We believe we have the correct format for this citation, since "Hmiel" is used as part of the sentence.*

Line 118: for what conditions (geomagnetic and solar) is the $P_{n, SLHL}^{Qtz}(0)$ defined?
*The Borchers et al. (2016) study that is the source of this value applies the scaling model of Lifton et al. (2014; reference already in our manuscript) to define this reference production rate for 2001-2010 mean solar modulation and geomagnetic field conditions. We will clarify this in the revised manuscript.*

Line 119: should the units be molecules (viz. 14CO) or atoms (viz. 14C)? Referring to to the text above, it should be atoms. Please check.
*We thank the referee for catching this, it should indeed be "atoms" -- we will update in the revised manuscript.*

Description after Eqs 2 and 3 are quoted from Hmiel et al. (2023) but this is not optimal since some important information is missing there as probably provided elsewhere in the cited paper. The authors are advised to describe the formulas, especially Eq.3, in full detail. In particular, it is not described how \beta(h) is obtained.
*We will revise the manuscript to have a more complete description of these equations*

Line 140: since the ablation exposes ancient ice to neutrons, the additional production of 14C by neutrons needs to be considered and possibly corrected for. From the subsequent narrative, I understand that this effect is neglected, but this is not clear.
*Our model does consider production from the neutron mechanism, but its contribution is negligible for samples at 6.85 m or deeper at Taylor Glacier. We will further clarify this in the revised manuscript*

Line 203+, also 270: while parameters R1 and L1 are described, it is unclear how they are used. Please provide a formula for that.

*These parameters were explained more completely in the cited Hmiel et al. (2023) study, but we will clarify further / provide any applicable formulas in the revised manuscript.*

Figure 4a: The Y-axis can be plotted logarithmically (optional).
*We would prefer to keep the linear scale, as very low values (which would be given more prominence on a logarithmic scale) are not measurable / negligible for the purposes of this study*

Lines 350-351: please remove quotation marks.
*We think the quotation marks are useful and appropriate here for clarity, as we are exactly reproducing text from a figure legend.*

The unnumbered equation in line 354 is unclear. I am ignorant of this but it doesn't look like the probability (e.g., can it be greater than unity if Delta a is small?). Please explain this formula and/or give a reference.
*We thank Dr. Usoskin for catching the fact that equations in Section 3.3 were unnumbered -- we will add equation numbers in the revised manuscript.*
*This particular equation is a probability density function (PDF) defined to be uniform between $a_{min}$ and $a_{max}$ and 0 elsewhere. It is normalized to 1 when integrating over all values of the rate of change a, and thus takes on the value $1/(a_{max} - a_{min}) = 1/\Delta a$ after normalization. It is a proper probability distribution: even though it is possible for $1/\Delta a > 1$, the integral of the PDF over all possible values of a is unity. We note that improper (non-normalizable) uniform priors are valid and are often used in Bayesian statistics (though not in this case). This is allowed as long as the marginal distribution using an improper prior is normalizable. We will explain this equation in more detail in the revised manuscript.*

Line 376: please check that the term "frequentist probability" is correctly used here.
*While the term is jargon, it is used correctly. We use it to distinguish the notion of probability as the expected outcome in many repeated measurements ("frequentist") from the interpretation of probability reflecting ignorance or prior information ("Bayesian"). In our sensitivity calculation, the p-value expresses the number of times we expect to measure $B_{01}$ as small or smaller than what we observe in our actual measurement, in the case that no temporal effects are present in the GCR flux. Since we have used a Bayesian statistic, the Bayes Factor, to perform the model comparison, we feel it is important to note that our sensitivity is a frequentist calculation.*

Lines 382-384 repeat what is said in lines ~330.
*That is correct and the repetition is intentional for clarity. Lines ~330 describe depth-14CO profile shapes on the figure, with the intention of giving the reader a visual introduction to how temporal variability in GCR flux could affect the shape of the profiles. Lines ~383 summarize the temporal variations trialed in the sensitivity analysis.*

Line 388: was the step-like increase at 3.5 ka or 3 ka as stated in line 331?
*For the sensitivity analysis (line 388), the transient step-like increase is at 3.5 ka. For the visual introduction to the effects of temporal variability (Figure 4b, line 331), this increase was at 3 ka, so both are correct.*

Line 405: lunar rocks can be also mentioned here.
*As the lunar rock evidence would already be added in the Introduction (as per Dr Usoskin's earlier comment), and as lunar rocks provide only an indirect indication that the GCR flux could have been constant, without quantifying uncertainties, we would prefer to not mention it again here.*

Line 411: measurements of d15N were not discussed in the text and appear out of the blue here. It needs to be removed or introduced somewhere.
*We will remove mention of d15N to avoid confusion*

---

## Author Response (AR1)

Referee comments in regular font
*Our responses in italics*

**RC1: ['Comment on egusphere-2023-3126'](), Anonymous Referee #1, 06 Feb 2024**

Petrenko et al. explore the potential of 14CO measurements in ice cores as a means to investigate the stability of the galactic cosmic ray (gcr) flux outside the heliosphere (the local interstellar spectrum, LIS). The rationale behind the approach is, that 14CO production in firn/ice below ~70 m is dominated by fast muons which are produced by gcr of very high energy (>100 GeV). Such high energy gcr are nearly unaffected by the helio- and geomagnetic fields. Thus, changes in 14CO production by fast muons can inform about changes in the flux of gcr outside the heliosphere, which is of importance for all studies using cosmogenic radionuclides (e.g., solar activity reconstructions) as the constancy of the LIS is an underlying assumption to all of them. The paper first reviews previous data and modelling results previously obtained from Greenland and Antarctic ice cores that allowed narrowing down the uncertainty of some of the required parameters for understanding 14CO-concentrations in ice. This lead up to the formulation of the method conceptualized here and to the identification of site characteristics required for testing it. Using the site of EPICA Dome C as an example, the authors employ a firn model and an ice-only model to demonstrate the expected importance of the different 14CO production mechanisms over depth and the effect of prescribed changes in the cosmic ray flux on 14CO concentrations in the ice.

Lastly, the authors test, under which scenario a change in the LIS could be detected from measurements. Owing to the large temporal averaging of the big samples and the large penetration depth of fast muons, these results indicate that short term changes in the LIS are unlikely to be detected using this method (or at least the changes have to be so big that they would also become obvious from simpler methods). But the method may be able to detect a linear increase of 4% of the GCR flux over the Holocene.

This paper is building upon the work of the same group of authors and it is another great addition to the portfolio of scientific questions that may be asked through gas-specific 14C-analyses in ice. The possibility to test the stability of the LIS over time is intriguing and the paper is well written and scientifically excellent.

Hence, I only have some minor comments that I will outline below.

*We thank the referee for their helpful review and address the individual comments below.*

Minor Comments:

14CO can only provide constraints on the stability of the LIS above 100 GeV (the energy required for deep production by fast muons). The production of 10Be on the other hand is mainly caused by primary protons below 10 GeV (because there are so much more). With respect to the possibility to use 14CO as a constraint for the assumption of a constant LIS in 10Be-based solar activity reconstructions: Over which energy range can changes in the LIS be assumed to be proportional?

*This is a good question. The answer depends on what we hypothesize as the origin of any time-dependent changes in the GCR/LIS flux. If a point-source transient such as a supernova produces a local enhancement in the GCR flux, then the time-dependent effect should be more pronounced at higher energies. This is because the diffusion length of cosmic rays in the galactic magnetic field scales with energy. A simple way to understand this is by considering the*

*gyroradii of protons in the galactic magnetic field; at 10 GeV, it is ten times smaller than at 100 GeV, and is much smaller than the distance to the nearest star.*

*Thus, at lower energies it is likely that magnetic fields in the local interstellar medium will dampen time-dependent effects observed above 100 GeV. The extent to which that occurs is difficult to estimate a priori, as it could depend strongly on random fluctuations in the turbulent component of the galactic magnetic field as well as the position of the source of GCRs. To answer the question, we would need to model several scenarios using the largest and smallest reasonable values of the galactic field. That could be the subject of a future analysis but would be outside the scope of this study.*

*On the other hand, there may be scenarios where an enhancement of the local GCR flux is not due to a single point source, but due to our entry into or exit from a region with an enhanced and isotropized GCR flux, such as a local bubble produced by many sources surrounding the Solar System. In that case, the overall normalization of the flux could be affected with less energy dependence than expected from a single point source of cosmic rays.*

*In general, however, it is true that a constant LIS/GCR flux at high energy is very likely to imply a constant flux below 10 GeV, while a time-dependent flux above 100 GeV could still be consistent with a constant flux at 10 GeV. We have added a brief paragraph in the Conclusions section of the revised manuscript to address this.*

L19: "GCR flux": I would replace this with the term "local interstellar spectrum (LIS)" as this is clearly defined to be outside the heliosphere and thus outside the influence of helio- or geomagnetic modulation. Please also check this for the remainder of the manuscript, as I think in most instances, it is the LIS you're referring to and not e.g., the GCR-flux into the atmosphere. *The use of the term "local interstellar spectrum (LIS)" is not one we have often seen used to refer to cosmic rays beyond the influence of the heliosphere, but we acknowledge this may be due to differences in the literature on low-energy and high-energy cosmic rays. When referring to galactic particles above several hundred GeV, the terms "GCR flux" and "GCR spectrum" are often used interchangeably in the literature, with the phrase "solar-modulated flux" referring to heliospheric effects on the spectrum (and observed flux at Earth) below 100 GeV/nucleon.*

*While we appreciate the referee's constructive suggestion to improve the clarity of the paper, we feel that "GCR flux" is the more standard term used for cosmic rays beyond the heliosphere at the energy range relevant to this study, and thus we prefer to keep that term.*

L27-28: "linear change/step increase": define for which duration
*We clarified this in the revised manuscript abstract*

L37: "14C abundance": Replace with 14C/12C
*We made this change in the revised manuscript*

L45: "solar GCR flux modulation": Consider changing to "solar modulation of the GCR flux", as I was for a second thinking you're referring to solar cosmic rays.
*We made this change in the revised manuscript*

L52: "suggest": replace with "assume" – I don't think the mentioned studies provide any results to suggest this
*We made the suggested change in the revised manuscript*

L143: The 14C half-life has been revised (https://doi.org/10.1016/j.nuclphysa.2003.11.001)
*We made this change in the revised manuscript (now on line 180)*

L275-280 (& figure 4): where does the increase in 14CO from shallow processes (dashed blue) come from? Is this only an increase in closed porosity?
*Yes, this is due to increase in closed porosity and 14CO in porosity being trapped into bubbles. Most of the 14CO that is produced in the shallow firn leaks out into open porosity, and a fraction of this 14CO diffuses down to the lock-in zone, where it is then trapped in closed porosity (bubbles). We clarified this in the revised manuscript (now around line 325).*

Figure 4: I suggest to add a panel to illustrate the applied production changes. These are now only mentioned in the text, described by age, while the 14CO is plotted on depth. It would be nice to get a visual overview over all of it.
*In the revised manuscript, we added a panel (Fig. 4c) showing the relative production rate vs time in the time-variable scenarios, and updated the figure caption accordingly.*

L402: "the assumption of GCR flux constancy": add "during the Holocene" (and change to LIS)
*We added "during the Holocene" in the revised manuscript (now on line 658); please see above regarding "LIS" vs "GCR"*

The manuscript by Petrenko et al. presents a programmatic paper presenting and quantifying a novel approach to measuring 14CO in ice as a proxy of the long-term flux of high-energy (>100 GeV) cosmic rays. It is shown that this method is feasible and can indeed be used to study the GCR consistency on the time scale of thousands of years. The authors describe the physics behind the approach very well and provide a quantitative assessment of the method's sensitivity to demonstrate that it is sufficient for the task. The authors also correctly specify the related challenges.
I found this work very important expectedly becoming a reference paper for the new method.
I am happy to recommend this manuscript for potential acceptance subject to a minor revision related to some clarifications in the methodological description as specified below.
*We thank Dr. Usoskin for his detailed and helpful review and address individual comments below*

A reader would benefit from a brief general description of how 14CO is measured in ice.
*We have added a paragraph at the end of Section 2.1 in the revised manuscript to provide this description*

Line 24: "insensitive" -- > "almost insensitive".
*We made this change in the revised manuscript*

Line 33: after "solar irradiance" a reference to Wu et al. (2018b, doi: 10.1051/0004-6361/201832956) can be added.
*We have added this reference*

Line 35: please add a reference to a review by Usoskin (2023, doi: 10.1007/s41116-023-00036-z) focused on the cosmogenic method for solar activity reconstructions.
*We have also added this reference*

Line 44: In addition to meteoritic studies, cosmogenic isotopes in lunar rocks can provide an estimate of the very long-term (mega-years) flux of cosmic rays (see, e.g., Poluianov et al., 2018, doi: 10.1051/0004-6361/201833561 and references therein). This method is free of geomagnetic shielding and uncertainties related to the orbit and erosion, but of course, is strongly affected by solar modulation. This can be briefly mentioned here in addition to the meteoritic data.
*We added a brief mention of this study to the discussion, starting on line 50*

Line 57: the statement about the isotropy of the GCR flux at the level of 1 permil needs clarification and a reference. The flux of GCR (in the GeV energy range) near Earth has a level of anisotropy of about 1% due to the orbital motion and diffusion+convection of particles by solar wind. Probably, the authors' statement is related to higher energies. A reference is needed.
*The reviewer is correct that we are referring to the GCR flux at 100 GeV and beyond. There are two anisotropic signals observable at these energies:*

1. *A dipolar anisotropy due to the Earth's orbit around the Sun, producing an annual modulation due to Earth's relative motion through the "wind" of the local GCR flux.*

2. *An approximately dipolar anisotropy (with statistically significant components on angular scales down to a few degrees), most likely due to cosmic-ray diffusion from unresolved sources of cosmic rays within roughly 1 kpc of the solar system. The amplitude of the dipole, projected onto the celestial equator, varies between $10^{-4}$ and $10^{-3}$ between several hundred GeV and 1 PeV. Above 1 PeV, to 10 EeV (the highest reported energies) the dipole increases to $10^{-2}$.*

*A technical detail is that the solar anisotropy is a contaminant of the galactic anisotropy, but it can be estimated and removed by transforming cosmic-ray arrival directions to the so-called "anti-sidereal" reference frame (details are available in Farley & Storey, Proc. Phys. Soc. 67:996, 1954, Guillian+ Phys. Rev. D75:062003, 2007, and Aartsen+ ApJ 826:220, 2016).*

*Starting on line 63 in the revised manuscript, we added relevant references in the text that document the 0.1% GCR anisotropy at TeV energies and clarified the energy range this applies for.*

Line 113: "Hmiel et al. (2023)" -- > "Hmiel et al., 2023)".
*We believe we have the correct format for this citation, since "Hmiel" is used as part of the sentence.*

Line 118: for what conditions (geomagnetic and solar) is the $P_{n, SLHL}^{Qtz}(0)$ defined?
*The Borchers et al. (2016) study that is the source of this value applies the scaling model of Lifton et al. (2014; reference already in our manuscript) to define this reference production rate for 2001-2010 mean solar modulation and geomagnetic field conditions. We clarified this in the revised manuscript (now on line 144).*

Line 119: should the units be molecules (viz. 14CO) or atoms (viz. 14C)? Referring to to the text above, it should be atoms. Please check.
*We thank the referee for catching this, it should indeed be "atoms" -- we updated this in the revised manuscript (now on line 144)*

Description after Eqs 2 and 3 are quoted from Hmiel et al. (2023) but this is not optimal since some important information is missing there as probably provided elsewhere in the cited paper. The authors are advised to describe the formulas, especially Eq.3, in full detail. In particular, it is not described how \beta(h) is obtained.
*We have removed the quotation marks and the Hmiel et al (2023) reference; as the Hmiel et al (2023) manuscript is in review in parallel to this one we think this is acceptable. Description of all terms in Equation 2 was already complete. We have added detail regarding \beta(h), including two additional equations.*

Line 140: since the ablation exposes ancient ice to neutrons, the additional production of 14C by neutrons needs to be considered and possibly corrected for. From the subsequent narrative, I understand that this effect is neglected, but this is not clear.
*Our model does consider production from the neutron mechanism, but its contribution is negligible for samples at 6.85 m or deeper at Taylor Glacier – this was already mentioned*

*briefly in the Taylor Glacier section. We clarified this further in the revised manuscript (on line 182).*

Line 203+, also 270: while parameters R1 and L1 are described, it is unclear how they are used. Please provide a formula for that.
*These parameters were explained in detail in the cited Hmiel et al. (2023) study, which also provided a figure schematic to describe how the model handles $^{14}C$ retention and loss in the firn. As Hmiel et al was the original study to develop this framework and is accessible on the Cryosphere website (in discussion form), we think that in this case referring the reader to Hmiel et al for more details is appropriate. However, we have added some further clarification of how R1 and L1 are handled in the model (starting on line 246 in the revised manuscript).*

Figure 4a: The Y-axis can be plotted logarithmically (optional).
*We would prefer to keep the linear scale, as very low values (which would be given more prominence on a logarithmic scale) are not measurable / negligible for the purposes of this study*

Lines 350-351: please remove quotation marks.
*We think the quotation marks are useful and appropriate here for clarity, as we are exactly reproducing text from a figure legend.*

The unnumbered equation in line 354 is unclear. I am ignorant of this but it doesn't look like the probability (e.g., can it be greater than unity if Delta a is small?). Please explain this formula and/or give a reference.
*We thank Dr. Usoskin for catching the fact that equations in Section 3.3 were unnumbered -- we added equation numbers in the revised manuscript.*
*This particular equation is a probability density function (PDF) defined to be uniform between $a_{min}$ and $a_{max}$ and 0 elsewhere. It is normalized to 1 when integrating over all values of the rate of change a, and thus takes on the value $1/(a_{max} - a_{min}) = 1/\Delta a$ after normalization. It is a proper probability distribution: even though it is possible for $1/\Delta a > 1$, the integral of the PDF over all possible values of a is unity. We note that improper (non-normalizable) uniform priors are valid and are often used in Bayesian statistics (though not in this case). This is allowed as long as the marginal distribution using an improper prior is normalizable. We revised the presentation of the equation and explained this equation in more detail in the revised manuscript.*

Line 376: please check that the term "frequentist probability" is correctly used here.
*While the term is jargon, it was used correctly in the original manuscript. We used it to distinguish the notion of probability as the expected outcome in many repeated measurements ("frequentist") from the interpretation of probability reflecting ignorance or prior information ("Bayesian"). In our sensitivity calculation, the p-value expresses the number of times we expect to measure $B_{01}$ as small or smaller than what we observe in our actual measurement, in the case that no temporal effects are present in the GCR flux.*
*However, since both the referee and the editor found this confusing, we have removed this term in the revised manuscript.*

Lines 382-384 repeat what is said in lines ~330.

*That is correct and the repetition is intentional for clarity. Lines ~330 described depth-14CO profile shapes on the figure, with the intention of giving the reader a visual introduction to how temporal variability in GCR flux could affect the shape of the profiles. Lines ~383 summarized the temporal variations trialed in the sensitivity analysis.*

Line 388: was the step-like increase at 3.5 ka or 3 ka as stated in line 331?
*For the sensitivity analysis (line 388 in original manuscript), the transient step-like increase is at 3.5 ka. For the visual introduction to the effects of temporal variability (Figure 4b, line 331 in original manuscript), this increase was at 3 ka, so both are correct.*

Line 405: lunar rocks can be also mentioned here.
*As the lunar rock evidence would already be added in the Introduction (as per Dr Usoskin's earlier comment), and as lunar rocks provide only an indirect indication that the GCR flux could have been constant, without quantifying uncertainties, we would prefer to not mention it again here.*

Line 411: measurements of d15N were not discussed in the text and appear out of the blue here. It needs to be removed or introduced somewhere.
*We removed the mention of d15N to avoid confusion*

**Additional revision request from the Editor (Hubertus Fischer):**
Apart from the changes you already indicated in your reply, I would ask you to put special emphasis on explaining the statistical approach in section 3.3 a bit more in detail. Not all TC readers will be familiar with this approach and a descriptive explanation of the equations in lines 344, 354 and 362 as well as of the jargon expression "frequentists" would be most helpful.
*We have revised Section 3.3 substantially (see marked-up manuscript, too many changes to list individually) to further clarify the equations and the overall statistical approach involving Bayes factor. As already mentioned above, we no longer use the term "frequentist".*